computational chemistry

radical-scavenging reaction, long-range correction, dispersion correction, density functional theory, hydrogen atom transfer

**Authors for correspondence:**
Febdian Rusydi
e-mail: rusydi@fst.unair.ac.id
Hermawan K. Dipojono
e-mail: dipojono@tf.itb.ac.id

This article has been edited by the Royal Society of Chemistry, including the commissioning, peer review process and editorial aspects up to the point of acceptance.

# The significance of long-range correction to the hydroperoxyl radical-scavenging reaction of trans-resveratrol and gnetin C

Vera Khoirunisa[2,4,5], Febdian Rusydi[1,2], Lusia S. P. Boli[2,5], Ira Puspitasari[2,3], Heni Rachmawati[6,7] and Hermawan K. Dipojono[5,7]

[1]Department of Physics, [2]Research Center for Quantum Engineering Design, and [3]Information System Study Program, Faculty of Science and Technology, Universitas Airlangga, Jl. Mulyorejo, Surabaya 60115, Indonesia
[4]Engineering Physics Study Program, Institut Teknologi Sumatera, Jl. Terusan Ryacudu, Lampung Selatan 35365, Indonesia
[5]Advanced Functional Materials Research Group, [6]School of Pharmacy, and [7]Research Center for Nanosciences and Nanotechnology, Institut Teknologi Bandung, Jl. Ganesha no. 10, Bandung 40132, Indonesia

VK, 0000-0002-5899-8462; FR, 0000-0002-7224-5731; LSPB, 0000-0002-6687-1488; IP, 0000-0001-5983-6257; HR, 0000-0003-1968-0002; HKD, 0000-0002-1391-3533

Density functional theory has been gaining popularity for studying the radical scavenging activity of antioxidants. However, only a few studies investigate the importance of calculation methods on the radical-scavenging reactions. In this study, we examined the significance of (i) the long-range correction on the coulombic interaction and (ii) the London dispersion correction to the hydroperoxyl radical-scavenging reaction of trans-resveratrol and gnetin C. We employed B3LYP, CAM-B3LYP, M06-2X exchange-correlation functionals and B3LYP with the D3 version of Grimme's dispersion in the calculations. The results showed that long-range correction on the coulombic interaction had a significant effect on the increase of reaction and activation energies. The increase was in line with the change of hydroperoxyl radical's orientation in the transition state structure. Meanwhile, the London dispersion correction only had a minor effect on the transition state structure, reaction energy and activation energy. Overall, long-range correction on the coulombic interaction had a significant impact on the radical-scavenging reaction.

＊

# 1. Introduction

Radical scavenging is an important property of antioxidants. It is the act of antioxidant to deactivate or remove free radicals to prevent oxidative damage in the biological system. A common radical scavenging example is the inhibition process of lipid peroxidation. In the inhibition process, an antioxidant scavenges the peroxyl radical to stop the chain reaction, which leads to lipid peroxidation. A phenolic antioxidant, such as resveratrol, is known to scavenge the peroxyl radical by donating its hydrogen atom [1]. The hydrogen donation can be affected by non-covalent interactions such as hydrogen bonding and steric repulsion in the system. Therefore, it is expected that non-covalent interactions significantly influence the activity of phenolic antioxidant [2].

The radical scavenging activity of antioxidants is widely studied theoretically using density functional theory (DFT) [1,3–16]. DFT has successfully predicted the antioxidant activity through thermodynamic quantities [1,17–19]. However, the limitation of exchange-correlation functionals in DFT for non-covalent interaction and barrier height calculations [20–22] are the challenges for studying the radicals scavenging reaction that leads to reaction kinetics and mechanism. Therefore, corrections to exchange-correlation functionals are needed to overcome the limitations.

One way to handle the limitation of exchange-correlation functionals in DFT is the long-range correction. It improves calculations by partitioning exchange interaction into two regions, Hartree–Fock exchange at long-range interaction and pure DFT at short-range interaction [23,24]. However, long-range correction cannot describe the correct asymptotic $R^{-6}$ potential for large intermolecular distances. The potential can be described by dispersion correction, which adds an empirical term to account for dispersion [25–27]. Another way to overcome the limitations is by applying the exchange-correlation functionals from Minnesota density functionals. M06-2X functional, one of the Minnesota density functionals, has been tested in many cases—it improved the accuracy for thermodynamic, kinetics and non-covalent parametric quantities of various simple chemical reactions [28,29].

In this study, we use four calculation methods for studying the hydroperoxyl radical-scavenging reaction of trans-resveratrol and gnetin C. We aim to examine the effect of long-range and dispersion correction on the transition state and activation energy of the two radical-scavenging reactions. We use B3LYP as a referenced functional since it is the most popular density functionals in chemistry [30] and has provided a good prediction in our previous studies [31–33]. We use a version of B3LYP that has been corrected using the Coulomb-attenuating method in CAM-B3LYP exchange-correlation functionals [23] and B3LYP with the D3 version of Grimme's dispersion [25] for performing long-range correction on the coulombic interaction and London dispersion correction, respectively. As a comparison, we also use M06-2X functional [29]. The two radical-scavenging reactions are the representative model for the inhibition process of lipid peroxidation by melinjo resveratrol.

# 2. Model and computational details

## 2.1. Reaction model

We modelled the radical-scavenging reaction based on the hydrogen atom transfer (HAT) mechanism, as shown in scheme 1. In the initial and final states (abbreviated to [in.] and [fi.], respectively), molecules were in the ground state. The reactants were ROH (an antioxidant agent) and $^{\bullet}$OOH (hydroperoxyl, a model of peroxyl radicals in general). We used two antioxidant agents from melinjo–trans-resveratrol (tR) and gnetin C (gC)—as shown in figure 1a,b respectively. We only considered one active site of ROH for the hydrogen donation, which was $4'-OH$ site, as shown in figure 1c. It was the lowest bond dissociation energy among other sites [34,35]. Between [in.] and [fi.], we considered one transition state [TS], where we assumed the activated complexes, [RO—H—OOH], were formed.

## 2.2. Activation and reaction energy calculations

We constructed the reaction progress in an energy level diagram for scheme 1. It allowed us to calculate the reaction energy $(\Delta G^{\circ})$ and the activation energy $(\Delta^{\ddagger} G^{\circ})$ directly in terms of the standard Gibbs free

$$\text{ROH} + {}^{\bullet}\text{OOH} \longrightarrow [\text{RO}\text{-}\text{-}\text{H}\text{-}\text{-}\text{OOH}] \longrightarrow \text{RO}^{\bullet} + \text{H}_2\text{O}_2$$

[in.]    [TS]    [fi.]

**Scheme 1.** The radical-scavenging reaction model, where [in.], [TS] and [fi.] are initial, transition and final states.

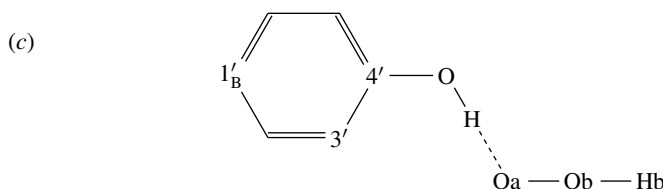

**Figure 1.** Molecular model for (*a*) trans-resveratrol (tR) in the ground state, (*b*) gnetin C (gC) in the ground state and (*c*) Ring B and hydroperoxyl in the [TS]. The nomenclature is used throughout the manuscript. The numbering of oxygen and hydrogen followed the numbering of its attaching carbon. The A, B, C and D indexes at the first carbon atom in each ring referred to phenyl ring's labels. The prime, dagger and double dagger symbols were for numbering ring B, C and D, respectively.

energy at 298.15 K. $\Delta G^{\circ}$ was the energy difference between [fi.] and [in.]

$$\Delta G^{\circ} = (G^{\circ}_{\text{RO}^{\bullet}} + G^{\circ}_{\text{H}_2\text{O}_2}) - (G^{\circ}_{\text{ROH}} + G^{\circ}_{{}^{\bullet}\text{OOH}}),\tag{2.1}$$

while $\Delta^{\ddagger}G^{\circ}$ was the energy difference between [TS] and [in.]

$$\Delta^{\ddagger}G^{\circ} = G^{\circ}_{\text{TS}} - (G^{\circ}_{\text{ROH}} + G^{\circ}_{{}^{\bullet}\text{OOH}}).\tag{2.2}$$

$G^{\circ}$ is the total electronic energy with a correction from Gibbs free energy.

**Table 1.** List of methods and their notation.

| | | |
|---|---|---|
| M1 | B3LYP | the reference throughout the analysis |
| M2 | B3LYP with GD3 | London dispersion correction |
| M3 | CAM-B3LYP | long-range correction on the coulombic interaction |
| M4 | M06-2X | parameterization and evaluation for non-covalent interactions |

## 2.3. DFT calculation set-up

We used DFT calculations for obtaining the geometry in the ground and the transition states. In addition to the DFT calculations, we coupled it with frequency calculations at 298.15 K to determine the Gibbs free energy correction. Furthermore, we used natural bond orbital (NBO) calculations for the charge population analysis.

While we only used one basis set, which was 6-31++G(d,p), we performed all calculations using three different exchange-correlational functionals (XCs), namely B3LYP, CAM-B3LYP and M06-2X. We also performed the calculations using B3LYP with D3 version of Grimme's dispersion (GD3). Therefore, we were able to study the effect of long-range correction on the coulombic interaction and London dispersion correction in the transition state. List of methods and their notation is shown in table 1.

The routine calculations were as follows. First, we performed the geometry optimization to obtain the most stable spin-state. We considered singlet, triplet and quintet spin state for molecules with an even number of electrons. As for molecules with an odd number of electrons, the doublet and quartet spin state were considered. Second, we used the most stable spin-state for further calculations to obtain the optimized geometry and energy of molecules in the ground and transition state. We obtained the transition state by tracking a particular vibrational mode that decreased along the designed pathway, as demonstrated in our previous study [36]. We began the geometry optimization by employing B3LYP. The optimized structures were re-optimized with B3LYP + GD3, CAM-B3LYP and M06-2X. The relevant activated complex structures were the ones with the vibration of H atom between the 4′—OH site and •OOH having the imaginary frequency. All calculations were done in the gas phase using Gaussian 09 software [37].

# 3. Results and discussions

## 3.1. The ground state structures

The optimization geometry calculations for trans-resveratrol and gnetin C using three XCs obtained spin-singlet state was the lowest in energy level. The energy difference between the singlet and triplet states was about 2.0 eV (trans-resveratrol) and 6.5 eV (gnetin C); while between the singlet and quintet states was about 6.2 eV (trans-resveratrol) and 10.4 eV (gnetin C). The differences are significant, which indicates that the spin-singlet state is very stable. The result agrees with most organic compounds that are stable in the spin-singlet state, with carbenes as the exception [38,39]. Therefore, we only considered the spin-singlet state for further calculations. As for hydroperoxyl, the spin-doublet was the ground state and the next spin state was a quintet with energy difference 2.8 eV on average.

Overall, the obtained ground state geometries of trans-resveratrol and hydroperoxyl were in good agreement with the experimental result, as shown in table 2(a.i)—(a.xii) and (b.xv)—(b.xvii). The discrepancies were less than 0.017 Å and 1.4 degrees, which were considered accurate for DFT calculations [30]. The higher discrepancies were for C4′—O bond length and Oa—Ob—Hb bond angle by M3 and M4. However, when we considered the experimental error, these values were still in the range. Therefore, all methods were capable to determine an accurate geometric structure for trans-resveratrol and hydroperoxyl. It implies we can use all methods for further calculations.

In detail, there was a significant difference in the dihedral angles of trans-resveratrol [table 2(a.xiii)]. The calculations obtained phenyl ring A and B were twisted, while experimental showed they were preferably planar. The NBO calculations determined that all hydrogens were positively charged (see electronic supplementary material, table S1); hence the coulombic repulsions of H2′—H$\alpha$ and H2—H$\beta$ were responsible for $D$(A, B). However, the coulombic repulsions were unlikely to play a dominant role in the experiment. As Zarychta *et al.* [40] reported, trans-resveratrol was prepared in crystal form, where one trans-resveratrol was surrounded by six others. Each trans-resveratrol formed hydrogen

**Table 2.** The selected geometric parameters of (a) trans-resveratrol and (b) hydroperoxyl in the ground state, the bond length ($R$, in Å), the bond angle ($A$, in degrees) and the dihedral angle ($D$, in degrees). Parameter (i)—(xii) and (xv)—(xvii) are the discrepancy from the experimental values. Parameter (xiii) is the difference between ring A and ring B calculated with the same method. Parameter (xiv) is the absolute value (without any reference).

|      | parameter | Expr. | M1 | M2 | M3 | M4 |
|------|-----------|-------|-----|-----|-----|-----|
| (a)  | trans-resveratrol | | | | | |
| (i)  | $R(3, 2)$ | 1.387 | +0.006 | +0.006 | +0.001 | +0.004 |
| (ii) | $R(2, 1)$ | 1.404 | +0.002 | +0.002 | −0.006 | −0.005 |
| (iii)| $R(1, \alpha)$ | 1.471 | −0.003 | −0.003 | −0.001 | 0.000 |
| (iv) | $R(\alpha, \beta)$ | 1.338 | +0.012 | +0.012 | +0.002 | +0.004 |
| (v)  | $R(\beta, 1')$ | 1.462 | +0.003 | +0.003 | +0.006 | +0.007 |
| (vi) | $R(1', 2')$ | 1.400 | +0.009 | +0.009 | +0.000 | +0.001 |
| (vii)| $R(2', 3')$ | 1.385 | +0.007 | +0.007 | +0.003 | +0.005 |
| (viii)| $R(5, 0)$ | 1.378 | −0.008 | −0.008 | −0.014 | −0.016 |
| (ix) | $R(4', 0)$ | 1.381 | −0.011 | −0.011 | −0.017 | −0.019 |
| (x)  | $A(4, 5, 6)$ | 121.1 | 0.0 | 0.0 | 0.0 | +0.1 |
| (xi) | $A(1, \alpha, \beta)$ | 126.0 | +0.8 | +0.6 | +0.2 | −0.7 |
| (xii)| $A(3', 4', 5')$ | 120.3 | −0.6 | −0.6 | −0.5 | −0.3 |
| (xiii)| $D(A, B)$ | 8.7 | 17.4 | 21.2 | 30.5 | 39.6 |
| (xiv)| $D(3', 4', 0, H)$ | 32.0 | 0.0 | 0.0 | 0.1 | 0.7 |
| (b)  | hydroperoxyl | | | | | |
| (xv) | $R(0a, 0b)$ | 1.335 | −0.001 | −0.001 | −0.014 | −0.023 |
| (xvi)| $R(0b, Hb)$ | 0.977 | +0.004 | +0.004 | +0.001 | −0.001 |
| (xvii)| $A(0a, 0b, Hb)$ | 104.1 | +1.4 | +1.4 | +1.7 | +1.7 |

Note: Experimental values: trans-resveratrol from [40]; hydroperoxyl from [41].

bonds with its six neighbour molecules through OH—O. The hydrogen bonds were predominant over the coulombic repulsions; hence the measured dihedral angle showed the rings were preferably planar.

We also showed the planarity of H4′ in term of the dihedral angle $D(3', 4', O, H)$ [table 2(a.xiv)]. This particular H atom would be interacting with ·OOH in the transition state. The experimental value showed that it was not planar. It was due to the aforementioned experimental condition. However, all methods obtained planar H4′ with respect to ring B. We shall recall this quantity later in the activation energy discussion.

We remarked that the dihedral angle calculation was sensitive to the calculation method. The comparison results among four calculation methods showed that dispersion (M2) and long-range correction (M3) increased the twisting $D(A, B)$. Both simultaneous corrections (M4) increased $D(A, B)$ even further. This trend was consistent for the case of gnetin C (see electronic supplementary material, table S2). The results suggest the long-range correction plays a dominant role in the twisting compared with the dispersion correction.

## 3.2. The transition state structures

Figure 2 shows the optimized structures in the [TS] of scheme 1 obtained from all calculation methods for both trans-resveratrol and gnetin C. All structures possessed a single imaginary frequency, which was the O4′—H—OOH vibration. The magnitudes of imaginary frequency were more than 1300 cm$^{-1}$ for trans-resveratrol and 1200 cm$^{-1}$ for gnetin C. These magnitudes were strong, which indicated that the O4′—H—OOH vibration encouraged the displacement of H4′. The displacement of H4′ can also be seen from the elongation of O4′—H bond length, which was about 0.140 Å (or, 15% longer than in its ground state). Meanwhile, the Oa—Ob bond of ·OOH was not significantly elongated (only about

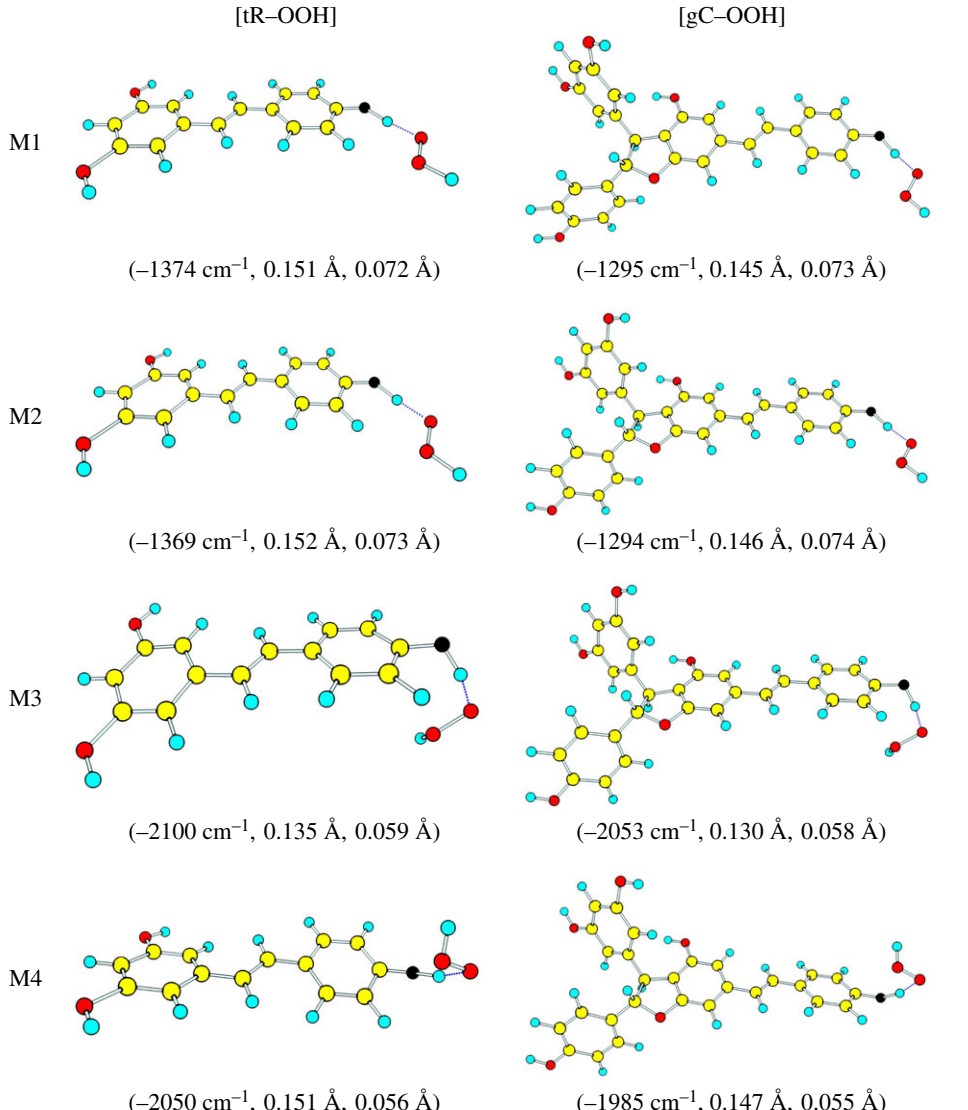

**Figure 2.** Optimized activated complex structures for trans-resveratrol and gnetin C with $^\bullet$OOH. Blue, red, yellow atoms represent H, O and C atoms. The black atom is O at the active site 4′, written as O4′ in the texts. The numbers in parentheses represent the required imaginary vibrational frequency for a transition state, the elongation of O4′—H and Oa—Ob bonds with respect to its ground state bond length.

5%). It means $^\bullet$OOH is attracting H4′. Therefore, the obtained structures are the activation complex of the radical-scavenging reaction in scheme 1.

The optimized [TS] structures revealed the different orientation of $^\bullet$OOH with respect to the ring B in the activated complexes. The presence of ring C and D did not contribute to the orientation, as the orientation was alike between [tR-OOH] and [gC-OOH]. However, the change in calculation methods altered the orientation. The significant alteration was obtained by M3 and M4, which implies that long-range correction plays a significant role in the orientation.

The orientation of $^\bullet$OOH in the activated complexes can be measured as a torsion angle of 3′— 4′— O—H, or $\phi$ (in degree). Table 3(a) shows the value of phi for all computational methods. Both M1 and M2 obtained phi was about zero, or $^\bullet$OOH was planar with respect to the ring B. However, $^\bullet$OOH was twisted up to 50 degrees according to M3 and M4 results. It implies that long-range correction was the reason for the twist. Therefore, the long-range correction plays a significant role both in the ground and the transition states of trans-resveratrol and gnetin C.

The origin of $^\bullet$OOH orientation is probably the same with the aforementioned ring A and B twisting origin in the ground state. The twisting presented after the long-range correction was introduced. Here, NBO calculations also determined that all the hydrogen atoms were positively charged, but both oxygen

**Table 3.** The difference of selected parameters of [RO—H—OOH] complex (figure 1c) from M1. (a) is the torsion angle (degree), (b–f) are the interatomic distance (Å), (g) is the bond angle (degree). For (h), the relative electronic energy (eV), M1 is set to be the reference.

| | parameter | ROH | M1 | M2 | M3 | M4 |
|---|---|---|---|---|---|---|
| (a) | $\phi(3', 4', 0, H)$ | tR | 0.0 | +1.0 | +43.4 | −52.8 |
| | | gC | 0.5 | +0.3 | +43.1 | −55.5 |
| (b) | $R(H4', Oa)$ | tR | 1.296 | −0.001 | −0.008 | −0.030 |
| | | gC | 1.307 | −0.002 | −0.010 | −0.034 |
| (c) | $R(H3', Hb)$ | tR | 3.234 | −0.041 | +0.274 | +0.384 |
| | | gC | 3.231 | −0.040 | +0.278 | +0.338 |
| (d) | $R(H3', Ob)$ | tR | 2.317 | −0.039 | +0.313 | +0.437 |
| | | gC | 2.314 | −0.038 | +0.319 | +0.460 |
| (e) | $R(Oa, Ob)$ | tR | 1.406 | 0.001 | −0.026 | −0.037 |
| | | gC | 1.406 | 0.001 | −0.027 | −0.039 |
| (f) | $R(Ob, Hb)$ | tR | 0.972 | 0.000 | +0.001 | 0.000 |
| | | gC | 0.971 | 0.000 | +0.001 | 0.000 |
| (g) | $A(Oa, Ob, Hb)$ | tR | 101.9 | 0.0 | +1.4 | +2.0 |
| | | gC | 101.9 | 0.0 | +1.5 | +2.1 |
| (h) | $E_{rel}$ | tR | 0 | −0.69 | +12.21 | +11.11 |
| | | gC | 0 | −1.65 | +22.30 | +19.15 |

Note: Negative value of $\phi$ means that O4'—H bond rotates in a clockwise direction.

atoms in •OOH were negatively charged. Therefore, coulombic interactions between the closest atoms in ring B and •OOH are the reason for the twisting. While Oa was attracted to H4', Hb was repelled by H3'. The effect of repulsion and attractions can be seen from the interatomic distance between these atoms. Table 3(b) and (c) show that H4'—Oa distances decreased while H3'—Hb distances increased after the long-range correction was introduced.

While the long-range correction determined the orientation of •OOH, the dispersion correction affected the interatomic distance [table 3(c) and (d)]. The latter contracted the interatomic distance of H—H and H—O by about 1.3% and 1.7%, respectively. The correction did not affect the covalent bond parameters [table 3(b), (e), (f) and (g)], which is reasonable since the dispersion only works in the non-covalent region. These results complemented the report by Grimme et al. [25]. They reported that the effect began to arise at about 2.0 Å for C—C interatomic distance. Meanwhile, the contraction of H—H and H—O showed a critical difference between M1 and M2. Both methods resulted in a planar •OOH's orientation, but the dispersion correction stabilized the activated complex, as shown in their electronic energy [table 3(h)]. The stability of the activated complex naturally affected the energy barrier so it may affect the kinetic study or even the reaction pathways.

## 3.3. The radical-scavenging reaction

Figure 3 shows the reaction progress of scheme 1 with the transition state described in the previous section. All methods predicted that the reaction was exergonic. The experiment demonstrated that this reaction was indeed exergonic by showing its observable antioxidant activity [42]. Even though the reaction occurred in the solution experimentally, other studies using DFT with M05-2X functional in aqueous solution also obtained exergonic [13,14]. Therefore, our results can be accountable for further analysis.

Even though all methods obtained an exergonic reaction for scheme 1, the dispersion (M2) and long-range (M3) correction led to a different result. Since the activated complex determined the product, the exergonic difference level was aligned with the orientation of •OOH: the more twisting, the less exergonic. Since the twisting was due to the long-range correction as we discussed previously [table 3(a–d)], it implies that long-range correction also affects a reaction's exergonic level.

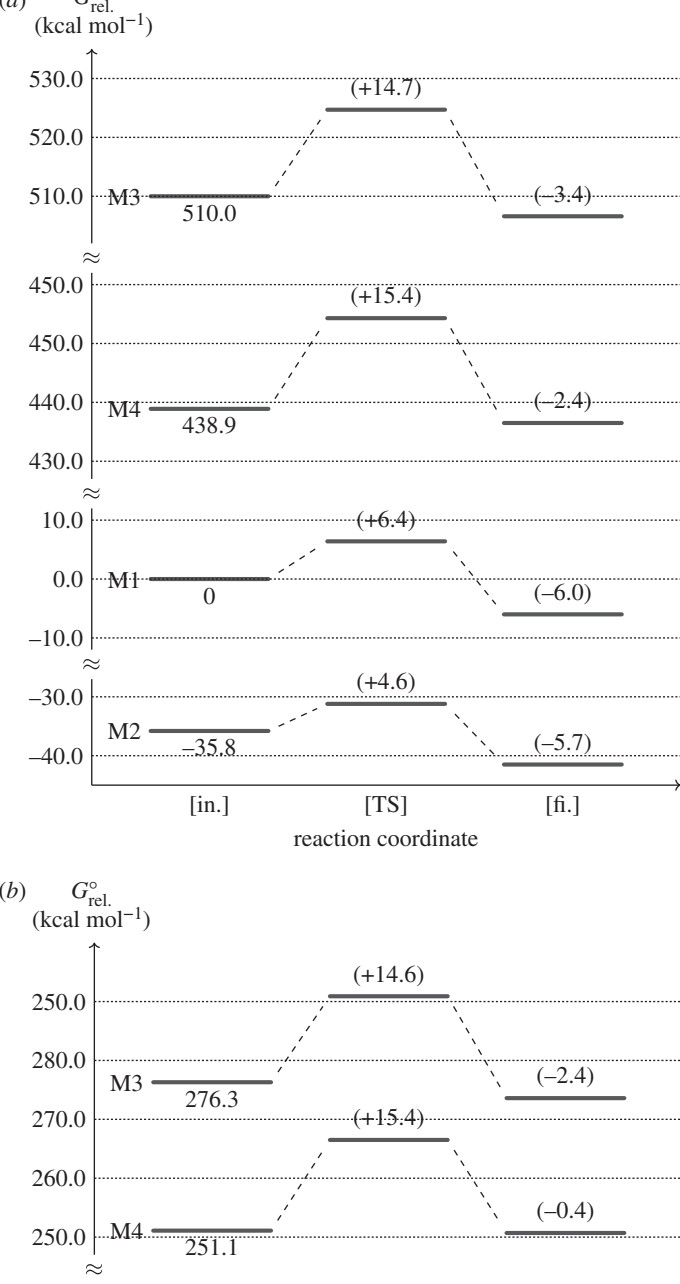

**Figure 3.** The radical-scavenging reaction of (*a*) trans-resveratrol and (*b*) gnetin C in an energy level diagram. The *y*-axis is the relative free Gibbs energy at room temperature ($G^\circ_{rel.}$) with the total energy of reactant calculated by M1 as the reference. For clarity, only $G^\circ$ of reactants are written and the parenthesized numbers are the Gibbs energy difference with respect to the reactant's total energy.

Dispersion (M2) and long-range correction (M3) also result in a different activation energy ($\Delta^\ddagger G^\circ$). When compared with M1, the former decreased $\Delta^\ddagger G^\circ$ by more than 20%, while the latter increased $\Delta^\ddagger G^\circ$ by more than 120%. The trend of the activation energy is similar to that of the activation

complex stability [table 3(h)]. It implies that the activated complex structure indeed determine the activation energy.

The increasing $\Delta^{\ddagger}G^{\circ}$ by the long-range correction was remarkable. Regarding the planarity difference of H4′ with respect to ring B between in the ground and the transition state, the former was plane [table 2(a.xiv)] and the latter was twisted [table 3(a)]. The results suggest that the increasing $\Delta^{\ddagger}G^{\circ}$ is due to the required energy to twist H4′ with respect to ring B.

Overall, the similarity in the higher $\Delta G^{\circ}$ and $\Delta^{\ddagger}G^{\circ}$ calculations by M3 and M4 is a noteworthy result. It appears that the similarity originates from the Hartree–Fock exchange functional contribution to the selected calculation methods. B3LYP functional (M1) contained 20% of the Hartree–Fock exchange functional [43], CAM-B3LYP (M3) contained 19% for short-range and 65% for long-range exchange [23] and M06-2X (M4) contained 54% [29]. Therefore, the exact exchange such as Hartree–Fock functional plays a significant role in this study. It supported the study by Zhao & Truhlar [29] that recommends the use of M06-2X for studying the thermodynamic, kinetics and non-covalent interactions of the main-group element. Specifically, this study validates the study by Chai & Head-Gordon [44] that showed the importance of long-range corrected hybrid functional in thermochemistry, kinetics and non-covalent interactions calculations.

# 4. Conclusion

We have reported the effect of long-range and dispersion correction on the hydroperoxyl radical-scavenging reaction of trans-resveratrol and gnetin C. We found that long-range correction on the coulombic interaction, which was included in CAM-B3LYP, showed significant effects on the reaction. The effects predicted by CAM-B3LYP were similar to that of M06-2X. Both CAM-B3LYP and M06-2X predicted higher reaction and activation energy (in terms of Gibbs free energy) than B3LYP. The increase was 2.6–3.6 kcal mol$^{-1}$ (trans-resveratrol) and 3.7–5.7 kcal mol$^{-1}$ (gnetin C) for the reaction energy, while for activation energy, the increase was up to 8 kcal mol$^{-1}$. We argued that the higher values of reaction and activation energy were due to hydroperoxyl radicals' twisted orientation in the transition state. Hydroperoxyl radical was twisted up to 50 degrees with respect to the phenyl ring attached to it. This twisted orientation of hydroperoxyl radical showed another similarity between CAM-B3LYP and M06-2X.

On the other hand, we noted that dispersion correction did not have a significant effect. B3LYP, without or with the Grimme's dispersion correction (GD3), obtained similar geometry and energy in the transition state. These results support other theoretical studies that reported the importance of long-range correction for the thermochemistry, kinetics and non-covalent interactions calculations. Therefore, our study verifies the significance of long-range correction in the hydroperoxyl radical-scavenging reaction of trans-resveratrol and gnetin C.

Data accessibility. The supporting data of this article has been uploaded as part of the electronic supplementary material.
Authors' contributions. V.K. carried out all simulations, participated in the design of the study, participated in data analysis and drafted the manuscript. F.R. conceived of the study, designed the methodology and critically revised the manuscript. L.S.P.B. and I.P. participated in data analysis. H.R. and H.K.D. critically revised the manuscript. All authors have read and agreed to the published version of the manuscript.
Competing interests. We declare we have no competing interests.
Funding. This work was supported by Direktorat Riset dan Pengabdian Masyarakat, Deputi Bidang Penguatan Riset dan Pengembangan Kementerian Riset dan Teknologi/Badan Riset dan Inovasi Nasional, Republik Indonesia under grant scheme Penelitian Dasar Unggulan Perguruan Tinggi (PDUPT) 2020 no. 798/UN3.14/PT/2020.
Acknowledgements. We thank Rizka Nur Fadilla (Universitas Airlangga) and Adhitya Gandaryus Saputro (Institut Teknologi Bandung) for the insightful discussions. V.K. particularly thanks Lembaga Pengelola Dana Pendidikan (LPDP) for the doctoral scholarship. All calculations using Gaussian 09 software are performed in the computer facility at Research Center for Nanoscience and Nanotechnology, Institut Teknologi Bandung, Indonesia and Global Education Center, National Institute of Technology, Akashi College, Japan.

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
