## [Peer Review File · Royal Society Open Science]

Review History

RSOS-201127.R0 (Original submission)

Review form: Reviewer 1

Is the manuscript scientifically sound in its present form?

No

Are the interpretations and conclusions justified by the results?

Yes

Is the language acceptable?

Yes

Do you have any ethical concerns with this paper?

Yes

Have you any concerns about statistical analyses in this paper?

No

Recommendation?

Accept with minor revision (please list in comments)

Comments to the Author(s)

Title: The significance of long-range correction to the hydroperoxyl radical-scavenging reaction of transresveratrol and gnetin C

Authors: Vera Khoirunisa, Febdian Rusydi, Lusia S.P. Boli, Ira Puspitasari, Heni Rachmawati, Hermawan K.

Dipojono

Manuscript ID: RSOS-201127

Summary:

The authors investigated the effects of different functionals and corrections on the radical scavenging reaction in hydroperoxyl by trans-resveratrol and gnetin C. The results showed that using different DFT based schemes yields different energetics and properties on the simulated reactions. Nonetheless, a significant impact of the long-range correction is observed on the energetics which is due to the modified orientation of the molecule's hydroperoxyl radical in the transition state. The authors presented interesting results and a systematic computational investigation. The results will provide an understanding of the mechanisms and a more accurate description of the reactions. The manuscript is recommended for publication but the authors should address the following comments and suggestions.

1. Discuss in the introduction the reason for choosing the molecules (hydroperoxyl, transresveratrol, gnetin C) in this study. What are their applications or what is/are the significance of these molecules?
2. Elaborate or provide the equation/s used in getting the energies in terms of the standard Gibbs free energy at 298 K.
3. Define clearly the meaning of positive and negative energies in Table 4. This will help the audience to understand the energy reaction profile. For instance, what is the physical meaning of the negative energy difference in Table 4b?
4. Are the differences in the energy values presented in Table 4 (especially for the activation energy) really that significant using the different functionals and/or corrections? Why? It will help if the authors show the energy vs the reaction path graph.
5. Given the obtained results, how to decide on which functional and/or correction should be used? If possible, cite experimental studies that will support the findings of this work.

Additional Comments:

1. The authors should identify the atoms in Figure 2.
2. The statement "See table 1 for calculation's orders" in page 5 may be confusing. Revise if necessary.

Review form: Reviewer 2

Is the manuscript scientifically sound in its present form?

No

Are the interpretations and conclusions justified by the results?

No

Is the language acceptable?

Yes

Do you have any ethical concerns with this paper?

Yes

Have you any concerns about statistical analyses in this paper?

No

Recommendation?

Major revision is needed (please make suggestions in comments)

Comments to the Author(s)

The authors report a computational chemistry study of the hydroperoxyl radical-scavenging reaction of trans-resveratrol and gnetin C. The main objective is to understand the role of long-range corrections and dispersion corrections in DFT methods on the structures and energetics of the reactions stated above.

Despite not working in radical-scavenging reaction I recognize this as an important class of systems in which computational chemistry method can contribute to their understanding. In this respect, it is important the choice of the DFT method to provide accurate predictions. This paper could be of interest to the community.

However, there are several aspects that should be considered by the authors before the manuscript could be considered for publication in RS Open Access:

1) Abstract: There is some confusion here in the authors' use of the terms "long-range" and "dispersion". I would consider "dispersion" a long-range force; I would re-word the abstract, and this part in particular, in order to avoid confusion to the reader.

2) Introduction: It would be useful for the reader to introduce what a radical-scavenging is.

3) Introduction: I don't like the first introductory paragraph. The sentence: "DFT offers many advantages in exploring the chemical properties of antioxidants molecules based on their quantum electronic structure" does not make sense because the same could be said by any material.

4) The introduction should be revised and extended considerably. In particular:

- Why is it critical to have a DFT method that includes long-range and dispersion corrections? The author mentions that B3LYP gave predictions (refs. 33-35)

- Why the two reactions used as benchmark? I

- Why the choice of these functionals? Four DFT methods have been chosen out of the hundreds available. In particular, I can understand the choice of CAM-B3LYP and B3LYP+D3. However, what is the connection with the M06-2X method?

5) Methods: The authors should explain in more detail what is the difference between the long-range and dispersion correction.

6) Results, Table 3, reaction energies. What are the values obtained with the B3LYP method?

7) Results, section 3.3. From this analysis, it is difficult to understand what method is right or wrong, especially for the reaction energies. The authors should find a real reference value such as higher-level calculations. This is an important aspect of the paper that the authors should

consider otherwise the manuscript is simply the comparison of the results obtained with four different DFT methods.

Decision letter (RSOS-201127.R0)

Dear Ms Khoirunisa:

Title: The significance of long-range correction to the hydroperoxyl radical-scavenging reaction of trans-resveratrol and gnetin C
Manuscript ID: RSOS-201127

The editor assigned to your manuscript has now received comments from reviewers. We would like you to revise your paper in accordance with the referee and Subject Editor suggestions which can be found below (not including confidential reports to the Editor). Please note this decision does not guarantee eventual acceptance.

Please submit your revised paper before 05-Sep-2020. Please note that the revision deadline will expire at 00.00am on this date. If we do not hear from you within this time then it will be assumed that the paper has been withdrawn. In exceptional circumstances, extensions may be possible if agreed with the Editorial Office in advance. We do not allow multiple rounds of revision so we urge you to make every effort to fully address all of the comments at this stage. If deemed necessary by the Editors, your manuscript will be sent back to one or more of the original reviewers for assessment. If the original reviewers are not available we may invite new reviewers.

On behalf of the Subject Editor Professor Anthony Stace and the Associate Editor Professor Kim Jelfs.

RSC Associate Editor:
Comments to the Author:
Please address carefully all of the reviewers' comments.

RSC Subject Editor:
Comments to the Author:
(There are no comments.)

Reviewers' Comments to Author:
Reviewer: 1

Comments to the Author(s)
Title: The significance of long-range correction to the hydroperoxyl radical-scavenging reaction of transresveratrol and gnetin C
Authors: Vera Khoirunisa, Febdian Rusydi, Lusia S.P. Boli, Ira Puspitasari, Heni Rachmawati, Hermawan K.
Dipojono
Manuscript ID: RSOS-201127

Summary:

The authors investigated the effects of different functionals and corrections on the radical scavenging reaction in hydroperoxyl by trans-resveratrol and gnetin C. The results showed that using different DFT based schemes yields different energetics and properties on the simulated reactions. Nonetheless, a significant impact of the long-range correction is observed on the energetics which is due to the modified orientation of the molecule's hydroperoxyl radical in the transition state. The authors presented interesting results and a systematic computational investigation. The results will provide an understanding of the mechanisms and a more accurate description of the reactions. The manuscript is recommended for publication but the authors should address the following comments and suggestions.

1. Discuss in the introduction the reason for choosing the molecules (hydroperoxyl, transresveratrol, gnetin C) in this study. What are their applications or what is/are the significance of these molecules?
2. Elaborate or provide the equation/s used in getting the energies in terms of the standard Gibbs free energy at 298 K.
3. Define clearly the meaning of positive and negative energies in Table 4. This will help the audience to understand the energy reaction profile. For instance, what is the physical meaning of the negative energy difference in Table 4b?
4. Are the differences in the energy values presented in Table 4 (especially for the activation energy) really that significant using the different functionals and/or corrections? Why? It will help if the authors show the energy vs the reaction path graph.

5. Given the obtained results, how to decide on which functional and/or correction should be used? If possible, cite experimental studies that will support the findings of this work.

Additional Comments:

1. The authors should identify the atoms in Figure 2.
2. The statement "See table 1 for calculation's orders" in page 5 may be confusing. Revise if necessary.

Reviewer: 2

Comments to the Author(s)

The authors report a computational chemistry study of the hydroperoxyl radical-scavenging reaction of trans-resveratrol and gnetin C. The main objective is to understand the role of long-range corrections and dispersion corrections in DFT methods on the structures and energetics of the reactions stated above.

Despite not working in radical-scavenging reaction I recognize this as an important class of systems in which computational chemistry method can contribute to their understanding. In this respect, it is important the choice of the DFT method to provide accurate predictions. This paper could be of interest to the community.

However, there are several aspects that should be considered by the authors before the manuscript could be considered for publication in RS Open Access:

1) Abstract: There is some confusion here in the authors' use of the terms "long-range" and "dispersion". I would consider "dispersion" a long-range force; I would re-word the abstract, and this part in particular, in order to avoid confusion to the reader.

2) Introduction: It would be useful for the reader to introduce what a radical-scavenging is.

3) Introduction: I don't like the first introductory paragraph. The sentence: "DFT offers many advantages in exploring the chemical properties of antioxidants molecules based on their quantum electronic structure" does not make sense because the same could be said by any material.

4) The introduction should be revised and extended considerably. In particular:

- Why is it critical to have a DFT method that includes long-range and dispersion corrections? The author mentions that B3LYP gave predictions (refs. 33-35)

- Why the two reactions used as benchmark? I

- Why the choice of these functionals? Four DFT methods have been chosen out of the hundreds available. In particular, I can understand the choice of CAM-B3LYP and B3LYP+D3. However, what is the connection with the M06-2X method?

5) Methods: The authors should explain in more detail what is the difference between the long-range and dispersion correction.

6) Results, Table 3, reaction energies. What are the values obtained with the B3LYP method?

7) Results, section 3.3. From this analysis, it is difficult to understand what method is right or wrong, especially for the reaction energies. The authors should find a real reference value such as higher-level calculations. This is an important aspect of the paper that the authors should

consider otherwise the manuscript is simply the comparison of the results obtained with four different DFT methods.

Author's Response to Decision Letter for (RSOS-201127.R0)

See Appendices A & B.

RSOS-201127.R1 (Revision)

Review form: Reviewer 1

Is the manuscript scientifically sound in its present form?

Yes

Are the interpretations and conclusions justified by the results?

Yes

Is the language acceptable?

Yes

Do you have any ethical concerns with this paper?

No

Have you any concerns about statistical analyses in this paper?

No

Recommendation?

Accept as is

Comments to the Author(s)

The authors made the necessary revisions and addressed the comments and questions. The manuscript is recommended for publication.

Decision letter (RSOS-201127.R1)

This year has been very difficult for everyone, and we want to take the opportunity to thank you for your continued support in 2020.

The Royal Society Open Science editorial office will be closed from the evening of Friday 18 December 2020 until Monday 4 January 2021. We will not be responding during this time. If you have received a deadline within this time period, please contact us as soon as possible to allow us to extend the deadline. If you receive any automated messages during this time asking you to meet a deadline, we offer apologies and invite you to respond after the festive period or during normal working hours.

With our best for a peaceful festive period and New Year, and we look forward to working with you in 2021.

Dear Ms Khoirunisa:

Title: The significance of long-range correction to the hydroperoxyl radical-scavenging reaction of trans-resveratrol and gnetin C

Manuscript ID: RSOS-201127.R1

It is a pleasure to accept your manuscript in its current form for publication in Royal Society Open Science. The chemistry content of Royal Society Open Science is published in collaboration with the Royal Society of Chemistry.

On behalf of the Subject Editor Professor Anthony Stace and the Associate Editor Professor Kim Jelfs.

RSC Associate Editor:
Comments to the Author:
(There are no comments.)

RSC Subject Editor:
Comments to the Author:
(There are no comments.)

Reviewer(s)' Comments to Author:
Reviewer: 1

Comments to the Author(s)
The authors made the necessary revisions and addressed the comments and questions. The manuscript is recommended for publication.

Appendix A

To: Reviewer 1

Subject: response and revision to the comments and questions

We are grateful for the reviewer's constructive comments on our manuscript. Here we respond to the reviewer's questions point by point. We mark the new sentences with the red font.

Sincerely,

on the behalf of the authors

Vera Khoirunisa

BEGIN

Question 1

Discuss in the introduction the reason for choosing the molecules (hydroperoxyl, trans-resveratrol, gnetin C) in this study. What are their applications or what is/are the significance of these molecules?

We choose trans-resveratrol and gnetin C because we are motivated to study the antioxidant activity of resveratrol in general and resveratrol from melinjo seed (melinjo resveratrol) in particular. To study the activity, we construct the radical-scavenging reaction model in the case of lipid peroxidation. In this reaction, we use $\bullet\text{OOH}$ as a general molecule model for a peroxy radical, the radical that has to be terminated to stop lipid peroxidation.

In alignment with other reviewer's requests, we revise the introduction majorly. We add a new paragraph as the first paragraph of "1.Introduction" section to introduce the radical scavenging reaction. The new paragraph is as follows.

In the manuscript

Radical scavenging is an important property of antioxidants. It is the act of antioxidant to deactivate or remove free radicals to prevent oxidative damage in the biological system. A common radical scavenging example is the inhibition process of lipid peroxidation. In the inhibition process, an antioxidant scavenges the peroxy radical to stop the chain reaction, which leads to lipid peroxidation. A phenolic antioxidant, such as resveratrol, is known to scavenge the peroxy radical by donating its hydrogen atom [1]. The hydrogen donation can be affected by non-covalent interactions such as hydrogen bonding and steric repulsion in the system. Therefore, it is expected that non-covalent interactions significantly influence the activity of phenolic antioxidant [2].

We also revise the last paragraph of “1.Introduction” section to state the reaction’s application involving trans-resveratrol, gnetin C, and hydroperoxy radical and the reason of choosing the methods. The paragraph is as follows. The revised sentences which relevant to the application are marked in bold.

In the manuscript

In this study, we use four calculation methods for studying the $\cdot\text{OOH}$ radical-scavenging reaction of trans-resveratrol and gnetin C. We aim to examine the effect of long-range and dispersion correction on the transition state and activation energy of the two radical-scavenging reactions. We use B3LYP as a referenced functional since it is the most popular density functionals in chemistry [30] and has provided a good prediction in our previous studies. [31-33] We use a version of B3LYP that has been corrected using the Coulomb-attenuating method in CAM-B3LYP exchange-correlation functionals [23] and B3LYP with the D3 version of Grimme’s dispersion [25] for performing long-range correction on the coulombic interaction and London dispersion correction, respectively. As a comparison, we also use M06-2X functional [29]. **The two radical-scavenging reactions are the representative model for the inhibition process of lipid peroxidation by melinjo resveratrol.**

Question 2

Elaborate or provide the equation/s used in getting the energies in terms of the standard Gibbs free energy at 298 K.

The equation for the standard Gibbs free energy is provided as follows.

In the manuscript

It allowed us to calculate the reaction energy (ΔG°) and the activation energy ($\Delta^\ddagger G^\circ$) directly in terms of the standard Gibbs free energy at 298.15 K. ΔG° was the energy difference between [fi.] and [in.]:

$$\Delta G^\circ = (G_{\text{RO}\cdot}^\circ + G_{\text{H}_2\text{O}_2}^\circ) - (G_{\text{ROH}}^\circ + G_{\text{OOH}}^\circ), \quad (1)$$

while $\Delta^\ddagger G^\circ$ was the energy difference between [TS] and [in.],

$$\Delta^\ddagger G^\circ = G_{\text{TS}}^\circ - (G_{\text{ROH}}^\circ + G_{\text{OOH}}^\circ). \quad (2)$$

G° is the total electronic energy with a correction from Gibbs free energy.

Question 3

Define clearly the meaning of positive and negative energies in Table 4. This will help the audience to understand the energy reaction profile. For instance, what is the physical meaning of the negative energy difference in Table 4b?

We have defined the physical meaning of positive and negative energies in Table 4 in our manuscript.

The energies in Table 4a is the reaction energy (ΔG°). It is the energy difference between the total energy of product and of reactant in term of Gibbs free energy.

Positive reaction energy means the reaction is not spontaneous (endergonic). Negative reaction energy means the reaction is spontaneous (exergonic).

The energies in Table 4b is the activation energy ($\Delta^\ddagger G^\circ$). It is the energy difference between activated complex and reactant energy in term of Gibbs free energy.

Positive activation energy means the reaction has a potential barrier. Negative reaction energy means the reaction is barrierless.

To clearly present the data to the readers, we decide to replace Table 4 with Figure 3. Figure 3 is the energy level of reactant, activated complex and product for all calculation methods. The y-axis is the relative energy, where we set the total energy of reactant calculated by M1 to be zero. The Figure 3 is as follows.

Question 4

Are the differences in the energy values presented in Table 4 (especially for the activation energy) really that significant using the different functionals and/or corrections? Why? It will help if the authors show the energy vs the reaction path graph.

Yes, the differences are significant in the activation energy when we use CAM-B3LYP (long-range correction) and M06-2X functional.

The difference in activation energy is about 8 kcal/mol. This amount of energy is significant to give a change on the vibrational mode of $\bullet\text{OOH}$. It requires 5.8 kcal/mol of energy to change the vibrational mode of $\bullet\text{OOH}$ from linear stretching mode (ν_1) to bending mode (ν_2). [1]

As it is suggested by the reviewer, we change Table 4 into energy level diagrams in Figure 3. We have shown Figure 3 earlier to answer Question 3.

Question 5

Given the obtained results, how to decide on which functional and/or correction should be used? If possible, cite experimental studies that will support the findings of this work.

Our study found the long-range coulombic interaction plays an important role in the radical scavenging reaction of resveratrol. The effect can be seen at the ground state structure, where the long-range coulombic interaction causes the twisting, as well as at the transition state structure (see Figure 2 in the manuscript) that directly impacts the energy barrier. Therefore, one needs to consider the exchange-correlation

(a) For trans-reveratrol scavenging

(b) For gnetin C scavenging

Figure 3: The reaction in Scheme 1 in an energy level diagram. The y-axis is the relative free Gibbs energy at room temperature (G°_{rel}) with the total energy of reactant calculated by M1 is the reference. For clarity, only G° of reactants are written and the parenthesized numbers are the ΔG° with respect to the reactant's total energy.

functionals with long-range columbic interaction formulae better than B3LYP has. CAM-B3LYP and M06-2X fit into this purpose.

In addition, our study extends the previous study reported by Iuga and Cordova-Gomez [2, 3] , where they used M02-5X to study the antioxidant activity of trans-resveratrol. However, they only used one exchange-correlation functional hence they did not explain what interaction plays the important role in the radical scavenging activity of trans-resveratrol with $\bullet\text{OOH}$.

Question 6

Additional Comments:

1. The authors should identify the atoms in Figure 2.
2. The statement “See table 1 for calculation’s orders” in page 5 may be confusing. Revise if necessary.

1. We take reviewer’s suggestion. We add a sentence in the caption of the Figure 2 to identify the atoms. The caption of Figure 2 is as follows.

In the manuscript

Figure 2. Optimized activated complex structures for trans-resveratrol and gnetin C with $\bullet\text{OOH}$. **Blue, red, yellow atoms represented H, O and C atoms.** The black atom is O at the active site 4' , written as O4' in the texts. The numbers in parentheses represent the required imaginary vibrational frequency for a transition state, the elongation of O4' –H and Oa – Ob bonds with respect to its ground state bond length.

2. We apologize for the confusion. We revise the sentence and move the sentence to the fifth paragraph of section “2. Model and Computational Details”. The revised sentence is as follows.

In the manuscript

While we only used one basis set, which was 6-31++G(d,p), we performed all calculations using three different exchange-correlational functionals (XCs), namely B3LYP, CAM-B3LYP, and M06-2X. We also performed the calculations using B3LYP with D3 version of Grimme's dispersion (GD3). Therefore, we were able to study the long-range and dispersion correction effect in the transition state. **List of methods and their notation is shown in table 1.**

References

- [1] W. M. Haynes 2014 CRC Handbook of Chemistry and Physics, 95th ed., CRC Press, Boca Rotan,Chp.9.
- [2] Iuga C, Alvarez-Idaboy JR, Russo N. 2012 Antioxidant Activity of trans-Resveratrol toward Hydroxyl and Hydroperoxyl Radicals: A Quantum Chemical and Computational Kinetics Study. *The Journal of Organic Chemistry* 77, 3868–3877.
- [3] Cordova-Gomez M, Galano A, Alvarez-Idaboy JR. 2013 Piceatannol, a better peroxyl radical scavenger than resveratrol. *RSC Advances* 3, 20209–20218.

Appendix B

To: Reviewer 2

Subject: response and revision to the comments and questions

We are grateful for the question and comments on our manuscript. We have revised the introduction majorly to address the reviewer's questions. We mark the new sentences with the red font.

Here we respond to the reviewer's questions point by point. We hope that our revised manuscript meets the reviewer's expectations.

Sincerely,

on the behalf of the authors

Vera Khoirunisa

BEGIN

Question 1

Abstract: There is some confusion here in the authors' use of the terms "long-range" and "dispersion". I would consider "dispersion" a long-range force; I would re-word the abstract, and this part in particular, in order to avoid confusion to the reader.

We can understand the confusion regarding the term long-range and dispersion. To differentiate the terms, we re-word and specify the terms in our revised manuscript. The changes are as follows.

In the manuscript

Density functional theory has been gaining popularity for studying the radical scavenging activity of antioxidants. However, only a few studies investigate the importance of calculation methods on the radical-scavenging reactions. In this study, we examined the significance of (1) the long-range correction on the coulombic interaction and (2) the London dispersion correction to the hydroperoxyl radical-scavenging reaction of trans-resveratrol and gnetin C. We employed B3LYP, CAM-B3LYP, M06-2X exchange-correlation functionals, and B3LYP with the D3 version of Grimme's dispersion in the calculations. The results showed that long-range correction on the coulombic interaction had a significant effect on the increase of reaction and activation energies. The increase was in line with the change of hydroperoxyl radical's orientation in the transition state structure. Meanwhile, the London dispersion correction only had a minor effect on the transition state structure, reaction energy, and activation energy. Overall, long-range correction on the coulombic interaction had a significant impact on the radical-scavenging reaction.

Question 2

Introduction: It would be useful for the reader to introduce what a radical-scavenging is.

We are glad the reviewer pointed out this issue. We agree that the introduction of radical-scavenging is useful for the reader. We add a new paragraph as the first paragraph in "1. Introduction" section to address this issue. The paragraph is as follows.

In the manuscript

Radical scavenging is an important property of antioxidants. It is the act of antioxidant to deactivate or remove free radicals to prevent oxidative damage in the biological system. A common radical scavenging example is the inhibition process of lipid peroxidation. In the inhibition process, an antioxidant scavenges the peroxy radical to stop the chain reaction, which leads to lipid peroxidation. A phenolic antioxidant, such as resveratrol, is known to scavenge the peroxy radical by donating its hydrogen atom [1]. The hydrogen donation can be affected by non-covalent interactions such as hydrogen bonding and steric repulsion in the system. Therefore, it is expected that non-covalent interactions significantly influence the activity of phenolic antioxidant [2].

Question 3

Introduction: I don't like the first introductory paragraph. The sentence: "DFT offers many advantages in exploring the chemical properties of antioxidants molecules based on their quantum electronic structure" does not make sense because the same could be said by any material.

We revise majorly the introduction to address all the reviewer's questions. Therefore, we no longer have this sentence in our revised manuscript.

Question 4

The introduction should be revised and extended considerably. In particular:

1. Why is it critical to have a DFT method that includes long-range and dispersion corrections? The author mentions that B3LYP gave predictions (refs. 33-35)
2. Why the two reactions used as benchmark?
3. Why the choice of these functionals? Four DFT methods have been chosen out of the hundreds available. In particular, I can understand the choice of CAM-B3LYP and B3LYP+D3. However, what is the connection with the M06-2X method?

We are aware that these questions arise since we do not clearly state our motivation and consideration of choosing the corrections, reactions, and methods. Therefore, we majorly revise our introduction to give a clear explanation.

Here is our answer to each question.

1. The interactions are classified into two: orbital (that leads to covalent bond) and dispersion interaction. Orbital interactions have short and long-range terms, while dispersion interactions only have long-range terms.

The mathematical formulation of interactions is defined by the exchange-correlation functional in DFT. B3LYP functional, for example, does not have a good description of the long range term of coulombic interaction as well as the dispersion interaction. While B3LYP is proven powerful for study ground state molecules, it gives some problems when we use it to study complex molecules, particularly in the transition state (reaction kinetic). That is why some new exchange-correlation functionals come to improve the results.

We revise the second paragraph of “1. Introduction” section to address this question. The paragraph is as follows.

In the manuscript

The radical scavenging activity of antioxidants is widely studied theoretically using density functional theory (DFT). [1,3-16] DFT has been successfully predicted the antioxidant activity through thermodynamic quantities. [1,17–19] However, the limitation of exchange-correlation functional in DFT for non-covalent interaction and barrier height calculations [20–22] can be challenges for studying the radicals scavenging reaction that leads to reaction kinetics and mechanism. Therefore, corrections to exchange-correlation functional are needed to overcome the limitations.

2. We use the two reactions as a benchmark because the two reactions are the reaction model for the inhibition process of lipid peroxidation by melinjo resveratrol.

We address this question in the last paragraph of “1. Introduction” section. The paragraph is as follows. The relevant sentences are marked in bold.

In the manuscript

In this study, we use four calculation methods for studying the $\cdot\text{OOH}$ radical-scavenging reaction of trans-resveratrol and gnetin C. We aim to examine the effect of long-range and dispersion correction on the transition state and activation energy of the two radical-scavenging reactions. We use B3LYP as a referenced functional since it is the most popular density functionals in chemistry [30] and has provided a good prediction in our previous studies. [31-33] We use a version of B3LYP that has been corrected using the Coulomb-attenuating method in CAM-B3LYP exchange-correlation functionals [23] and B3LYP with the D3 version of Grimme's dispersion [25] for performing long-range and dispersion correction, respectively. As a comparison, we also use M06-2X functional [29]. **The two radical-scavenging reactions are the representative model for the inhibition process of lipid peroxidation by melinjo resveratrol.**

3. M06-2X is a hybrid functional designed for the main group thermochemistry, barrier heights, and non-covalent interactions. [2, 3] Unlike CAM-B3LYP (long-range correction) and B3LYP+GD3 (dispersion correction), M06-2X accounts for non-covalent interactions implicitly by parameterization and has an improved accuracy to describe such a system. Therefore, we use M06-2X in our work.

The connection of choosing M06-2X with other functionals is explained in the third paragraph of the "1. Introduction" section. The paragraph is as follows. The explanation about M06-2X is marked in blue font.

In the manuscript

One to handle the limitations is long-range correction. It improves calculations by partitioning exchange interaction into two regions, Hartree-Fock exchange at long-range interaction and pure DFT at short-range interaction. [23,24] However, long-range correction cannot describe the correct asymptotic R^{-6} potential for large intermolecular distances. The potential can be described by dispersion correction, which adds an empirical term to account dispersion. [25–27] Another way to overcome the limitations is by applying the exchange-correlation functionals from Minnesota density functionals. M06-2X functional, one of Minnesota density functionals, has been tested in many cases—it improved the accuracy for thermodynamic, kinetics, and non-covalent parametric quantities of various simple chemical reactions. [28,29]

Question 5

Methods: The authors should explain in more detail what is the difference between the long-range and dispersion correction.

We understand the reviewer's concern. To maintain the story flow of the manuscript, we explain the difference between the long-range and dispersion correction in the third paragraph of "1. Introduction". The paragraph is as follows.

In the manuscript

One way to handle DFT limitations is long-range correction. It improves calculations by partitioning exchange interaction into two regions, Hartree-Fock exchange at long-range interaction and pure DFT at short-range interaction. [23,24] However, long-range correction cannot describe the correct asymptotic R^{-6} potential for large intermolecular distances. The potential can be describe by dispersion correction, which adds an empirical term to account dispersion. [25–27] Another way to overcome the limitations is by applying the exchange-correlation functionals from Minnesota density functionals. M06-2X functional, one of Minnesota density functionals, has been tested in many cases—it improved the accuracy for thermodynamic, kinetics, and non-covalent parametric quantities of various simple chemical reactions. [28,29]

Question 6

Results, Table 3, reaction energies. What are the values obtained with the B3LYP method?

There is no value of reaction energy obtained with the B3LYP method in Table 3. The only energy parameter in Table 3 is the relative electronic energy of [RO–H–OOH] complex, E_r . The [RO–H–OOH] complex energy obtained by M1 is used as a reference. Therefore, the energy is set to be zero.

To avoid misleading the reader, we decide to change E_r into E_{rel} in Table 3.

Question 7

Results, section 3.3. From this analysis, it is difficult to understand what method is right or wrong, especially for the reaction energies. The authors should find a real reference value such as higher-level calculations. This is an important aspect of the paper that the authors should consider otherwise the manuscript is simply the comparison of the results obtained with four different DFT methods.

We appreciate the reviewer's opinion. However, in this case, we have a different point of view.

To the best of our knowledge, there is no real reference value for the reaction and activation energy of the radical scavenging reaction of trans-resveratrol and OOH, both from experimental results and from higher-level calculation results.

Furthermore, this study is part of a continuous kinetic study. For the current stage, we focus on studying the interaction that can not be ignored in the radical scavenging activity of resveratrol, instead of obtaining accuracy in the activation and reaction energy. This study is only possible when we do a comparative study with four different DFT methods. We will continue our study to obtain accuracy in the kinetics of the radical scavenging reaction for future stage.

As a choice of functionals, we recommend exchange-correlation that incorporates the correction for the coulombic interaction in the long range region. CAM-B3LYP and M06-2X fits into this category.

END

References

- [1] Foresman JB, Frisch A. 2015 Exploring Chemistry with Electronic Structure Methods, 3rd ed., Gaussian, Inc.: Wallingford, CT.
- [2] La Rocca V, Malvina R, Ringeissen S, Gomar J, Frantz MC, Ngom S, Adamo C. 2016 Benchmarking the DFT methodology for assessing antioxidant-related properties: quercetin and edaravone as case studies. *Journal of Molecular Modeling* 22.
- [3] Zhao Y, Truhlar DG. 2008 The M06 suite of density functionals for main group thermochemistry, thermochemical kinetics, noncovalent interactions, excited states, and transition elements: two new functionals and systematic testing of four M06-class functionals and 12 other functionals. *Theoretical Chemistry Accounts* 120, 215–241